# Exploring the experiences of women of African, Caribbean and Mixed heritages to inform a music-based intervention for perinatal mental health in South East London: A qualitative study

Lottie Anstee[ID][1,*☉], Juliet Firth[2☉], Toyin Adeyinka[3], Katie Rose M. Sanfilippo[ID][4], Malik B. Jeng[5], Lauren Stewart[ID][1]

1 School of Psychology, University of Roehampton, London, United Kingdom, 2 Department of Psychology, Goldsmiths, University of London, London, United Kingdom, 3 South London and Maudsley NHS Foundation Trust, London, United Kingdom, 4 School of Health and Medical Sciences, City St George's, University of London, London, United Kingdom, 5 Yaram Arts, London, United Kingdom

☉ These authors contributed equally to this work and share first authorship.

* ansteel@roehampton.ac.uk

## Abstract

Women of Global Majority ethnicities have an increased risk of developing and sustaining perinatal mental health problems in the UK. This is partially explained by the ethnic inequalities experienced at an individual, societal and systemic level. Previous research highlights the benefits of engaging with participatory music-based interventions to alleviate symptoms of postnatal depression, stress and anxiety, but current provision lacks cultural inclusivity. This qualitative study focuses on fourteen women of African, Caribbean and Mixed heritages living in South East London to explore how their perinatal experiences, coping strategies and preferences regarding music-based support could inform a future culturally inclusive perinatal mental health participatory music intervention. Participants took part in either an online focus group or interview, led by a local community leader. An inductive reflexive thematic analysis identified four overarching themes: (1) supportive mechanisms during the perinatal period, (2) the overwhelming pressures and expectations of motherhood, (3) systemic barriers to accessing perinatal mental healthcare and (4) suggestions for future perinatal mental health music-based support groups. This study reveals the individual experiences of the perinatal period for women of African, Caribbean and Mixed heritages, exploring themes of sociocultural pressures, barriers to care and individual activities used to support mental health. The sociocultural, logistical and musical considerations outlined in this study highlight gaps in current community provision and offer practical suggestions for facilitating inclusive music-based interventions for perinatal mental health in South East London.

**Data availability statement:** All relevant data from the focus group and interview transcripts are available either in the manuscript or in S1 Table. In compliance with the ethical approval received from the Ethics Committee of the Psychology Department at Goldsmiths, University of London, full focus group and interview transcripts cannot be publicly shared, as participants did not consent to this and it would compromise participant confidentiality.

**Funding:** This work was supported by a grant from The Baring Foundation awarded to MBJ (grant number: 20220721). LA was supported by a South and East Network for Social Sciences (SENSS) Doctoral Training Partnership from the Economic and Social Research Council (ESRC). The funders had no role in study design, data collection and analysis, decision to publish or preparation of the manuscript.

**Competing interests:** The authors have declared that no competing interests exist.

## Introduction

One in five women in the United Kingdom (UK) will experience a mental health problem during the perinatal period [1], which encompasses pregnancy and up to a year post-birth. Poor perinatal mental health has been associated with several adverse effects, including mother-infant bonding difficulties [2] and long-term infant developmental complications [3–6]. Additionally, perinatal mental health problems are likely to worsen when left unrecognised and untreated, with suicide remaining the most common cause of death for mothers during the first year after birth [7–9]. However, women will often underplay their symptoms and not seek support due to various healthcare barriers [10], such as healthcare professionals dismissing concerns or normalising symptoms, leading to low diagnoses and treatment rates in the UK [3,11].

Various perinatal health inequalities have been identified in relation to ethnicity, socioeconomic background, age and multimorbidity [12,13], including an increased maternal mortality rate among women of Global Majority ethnicities [12]. Here, Global Majority refers to those who identify as Black, African, Asian, Brown, Mixed-heritage and indigenous [14,15]. Women of Global Majority ethnicities are less likely to be diagnosed or treated for perinatal mental health problems and more likely to experience limited access to and quality of care [16]. Evidence emphasises the systemic nature of ethnicity-related inequalities in statutory mental health services due to "monocultural and reductionist frameworks of assessment and treatment" and "direct experience of racist practice" [17, p.1]. These wider social and cultural contexts are reflected in and interact with the individual experiences of Global Majority women, where prevalent barriers include language and communication challenges, stigma, cultural expectations and family influences [18–21].

Research has suggested several approaches to improve the cultural sensitivity of healthcare services more generally, including person-centred care and cultural awareness training [22]. However, the systemic nature of the identified barriers and the longstanding mistrust of healthcare services among Global Majority communities [23] necessitates the expansion of perinatal support into other settings, such as community and voluntary sector organisations. Community-based care is designed to support health outside of statutory healthcare facilities, using local resources and networks, and may offer more space for directly developing interventions with women of Global Majority ethnicities to address their specific cultural and socioeconomic needs. This aligns with a recent call from the World Health Organisation [24] to focus on "evidence based, cost effective, and human rights oriented mental health and social care services in community-based settings for early identification and management of maternal mental disorders". In the UK, the 10 Year Health Plan for England [25] also explores the need to improve the accessibility of healthcare by engaging with communities, including through integrated neighbourhood teams, which aim to connect residents with various healthcare, social care and voluntary organisations tailored to their health needs.

Prior research suggests that co-locating mental health interventions within community settings could play a role in reducing health inequalities by improving the provision of holistic and person-centred support, increasing the accessibility of care and

providing psychologically safe environments separated from clinical services [26]. Community-based interventions could provide a useful avenue for peer support, which has been shown to reduce isolation and increase affirmation through shared experiences of motherhood [27]. Furthermore, these interventions could reduce stigma by being universally offered to all perinatal women, whether they are seeking a preventative support option, treatment for a perinatal mental health problem or a step-down intervention to sustain recovery. However, there also remains prevalent racial and ethnic disparities in access to community-based perinatal mental health interventions [28]. This emphasises the importance of working closely with women of Global Majority ethnicities to design and deliver novel, culturally informed community perinatal mental health interventions [29].

Participatory music-based interventions, where participants actively engage in musical activities, could represent a novel community-based perinatal support mechanism with the potential to address some of the persistent inequities experienced by Global Majority women. These interventions can foster social connections, self-development and enhanced mood regulation in diverse contexts [30]. Additionally, arts engagement has been theorised to address mental health inequities through a contextualised approach across individual, provider and system levels [31]. Therefore, participatory music-based interventions could provide an inclusive and cost-effective universal perinatal support mechanism applicable across the spectrum of mental health problems and across participants from diverse cultural backgrounds [32]. Several studies on participatory music-based interventions have found them to be effective for reducing symptoms of postnatal depression, anxiety and stress [33,34]. These interventions may also support women's emotional needs, increase social connectedness and equip women with musical skills to use in their daily lives [35–37]. While some research has been conducted with women of Global Majority ethnicities in other countries, including The Gambia [38], there has been little consideration of how inclusive perinatal mental health music-based interventions can be developed with and for diverse women in the UK.

This qualitative study explores the experiences of perinatal mental health support amongst women living in South East London and how these could inform the co-development of a culturally inclusive community music-based intervention. We focus specifically on women of African, Caribbean and Mixed heritages, who experience a higher risk of maternal morbidity than women from other Global Majority ethnicities [12,13]. Our approach acknowledges that women from different Global Majority ethnicities have distinct perinatal mental health experiences. This study explores the experiences of women of African, Caribbean and Mixed heritages to develop an understanding of their specific perinatal mental health needs, which could be used to tailor future community support mechanisms using a culturally informed approach. In this study, we address three key research questions:

1. What are the lived experiences of perinatal mental health among a sample of women of African, Caribbean and Mixed heritages living in South East London?

2. What musical and other support mechanisms are used by these women during the perinatal period?

3. What are their suggestions and preferences regarding a future music-based activity for perinatal mental health?

In doing so, our study addresses a significant gap within current psychological research around the needs of Global Majority women during the perinatal period and possible supportive mechanisms, including music, which is pertinent due to the increased maternal death and morbidity rates for this population.

## Methodology

### Ethics statement

This study received ethical approval from the Ethics Committee of the Psychology Department at Goldsmiths, University of London, on June 29, 2023. Participants provided electronic written informed consent after reviewing a participant information sheet, which included information on data confidentiality and right to withdraw. Participants were given a voucher

to reimburse them for their time. This study follows the COnsolidated criteria for REporting Qualitative research (COREQ) checklist [39].

## Setting

The setting for this study, South East London, encompasses the London boroughs of Southwark, Lambeth, Lewisham, Greenwich, Bexley, Bromley and their surrounding areas. The boroughs within South East London have a high level of ethnic diversity. For example, in a survey of residents from Southwark and Lambeth, 21.9% of participants identified as Black Caribbean or Black African [40]. Additionally, census data from 2021 reported that 26.8% of Lewisham residents identified as Black, Black British, Caribbean or African [41].

## Design

This study used a qualitative research design to centralise the voices of the participants and gain rich insights into their perinatal experiences. It describes the inductive reflexive thematic analysis of eight focus groups and interviews conducted with women of African, Caribbean and Mixed heritages living in South East London.

## Participants

Fourteen participants were recruited between July 1, 2023 and September 1, 2024. The inclusion criteria were: women who have given birth to at least one infant, are able to speak and understand English, are aged 18 years or older, live in South East London and identify as being from an African, Caribbean or Mixed heritage background. Women were included who identified with one of the following ethnicities: Caribbean, African, White and Black Caribbean, White and Black African or any other Black, Black British or Caribbean background. The recruitment was conducted by a local community leader from South East London (TA), who identifies as being from an African, Caribbean and Mixed heritage background and has experience supporting perinatal women in the community. Potential participants were contacted via text message or email to inform them about the study. Recruitment was initially conducted purposively, with TA contacting potential participants from existing connections, and later through snowball sampling to utilise the networks of participants. This sampling method ensured efficient recruitment of participants who matched the specific demographic required for our study. In addition, their connection to TA meant such participants might feel more comfortable talking about their experiences of the perinatal period. No interviewees refused to participate or wished to withdraw.

## Materials

A semi-structured focus group and interview guide was created by the authors in consultation with professionals from a local voluntary sector organisation and through reviewing previous literature. The focus groups and interviews offered a space for participants to elaborate on their lived experiences of the perinatal period and their perspectives on perinatal support mechanisms, including the potential uses of music. Questions centred on the following topics: (1) the types and impact of perinatal stresses, (2) the factors contributing to perinatal mental health problems, (3) existing perinatal support mechanisms, (4) the potential uses of musical activities and (5) suggestions for perinatal mental health music-based support groups. We invited women to retrospectively reflect on their experiences of the perinatal period and none of the women were in the perinatal period at the time of study. Participants were made aware of the study's intention to explore ideas around culturally relevant interventions on the information sheet and at the beginning of the discussion. The semi-structured nature of the focus groups and interviews allowed similar questions to be asked across the participant group to align with the research questions and aims of the study. Nevertheless, the question guide was used flexibly alongside follow-up questions to delve into participants' responses and enable women to share the meaningful aspects of their lived experiences. TA also tailored the questions to the different dynamics of the interview and focus group settings,

including more guided reflection on the commonalities across participants' experiences in the focus groups. The guiding questions can be found in Table 1, labelled according to their corresponding topics listed above.

## Procedure

Each participant took part in either a focus group or an individual interview, which were all led by TA. Eight focus groups and interviews were conducted between September 2023 and September 2024, with an average duration of 53 minutes (range = 19–88 minutes). Table 2 details the number assigned to each focus group and interview, alongside the number of participants who took part in each. Focus groups and interviews were conducted via Microsoft Teams and video recorded with automatic transcription following the consent of each participant. Participants were aware of the reasons behind the research and some had existing connections with the interviewer.

## Analysis

A reflexive thematic analysis was chosen to consider how knowledge is constructed through the salient features of participants' responses and acknowledge the influence of researcher positionality [42–44]. The analysis was completed by two research assistants, JF and LA, who both have training and experience in qualitative methods. Both authors regularly discussed how their individual subjectivities and opinions on the benefits of perinatal music influenced their interpretations,

**Table 1. Semi-structured focus group and interview questions.**

| Question |
| --- |
| 1. What sorts of stresses might women face during the perinatal period? (1) |
| 2. What are the possible effects of mental health issues during and after pregnancy? How severe can it be (for mother and baby)? (1) |
| 3. What factors do you think contribute to mental health issues during the perinatal period? (2) |
| 4. What kinds of things can women do to help themselves if they experience mental health issues during the perinatal period? (3) |
| 5. Do you already use music in any way to support your mental health during the perinatal period? In what way(s)? (Perhaps give some examples?) (3/4) |
| 6. What kind(s) of musical activity/activities do you think would be beneficial (if any) for women dealing with mental health struggles during the perinatal period? (4) |
| 7. Do you think a support group that involves engaging in musical activities together would be beneficial for mental wellbeing during the perinatal period? And why? Please describe what activities you think would work well. (5) |
| 8. What would be your preferred way of attending such an activity (for example, timing and location)? What barriers might you face to attending? (5) |

**Table 2. Characteristics of the focus groups and interviews.**

| Focus group/interview number | Number of participants |
| --- | --- |
| Focus Group 1 | 3 |
| Focus Group 2 | 3 |
| Focus Group 3 | 2 |
| Focus Group 4 | 2 |
| Interview 1 | 1 |
| Interview 2 | 1 |
| Interview 3 | 1 |
| Interview 4 | 1 |

following the principles of reflexivity [43]. Acknowledging these subjectivities enabled them to purposefully incorporate a diverse range of preferences regarding music interventions for perinatal mental health, including negative perspectives, to centre the voices of the participants. Additionally, both analysts acknowledged their privileged positions as White women and recognised the importance of exploring the influence of their background on each analytical step through reflexive journaling. JF and LA discussed all evolving interpretations, potential cultural biases and clarifications with TA to ensure that each theme meaningfully and accurately represented the voices and experiences of women of African, Caribbean and Mixed heritages. The analysis was conducted according to the philosophical underpinning of constructivism, which acknowledges that each individual's reality is shaped by both subjective perspective and past lived experiences in interplay with social interactions [45,46]. This theoretical basis enabled the analysts to consider individual meanings of each participant's experiences as a reflection of the wider social and cultural contexts within which their realities were formed. The constructivist lens was also used to explore inequalities at play on a personal, social and systemic level for each participant [47]. The design choice to have a local community leader with shared Global Majority heritage to lead the focus groups and interviews enabled participants to discuss past experiences of the perinatal period while sharing ideas related to broader systemic inequalities; through a constructivist lens, analysis considered the way that knowledge was thus co-created within the focus group and interview spaces [48].

Initially, the automatic transcriptions were edited using the recording to check for accuracy. Line-by-line coding was carried out by each analyst independently, before collaborating to iteratively refine the codes and themes. Transcripts were read several times for familiarisation, enabling the analysts to document preliminary reflexive and analytical thoughts. The two analysts individually systematically coded each transcript using NVivo, identifying salient and meaningful features of each participant's narrative through implicit and explicit interpretations of the data. The process was inductive to acknowledge the exploratory nature of the analysis and centre the lived experience voices of participants. The coding was also conducted iteratively and flexibly to incorporate additional interpretations as they were considered. The two analysts then collaboratively developed a map of themes and subthemes by collating the codes into coherent meaning-united interpretations. The themes were developed through aggregation of codes with shared meanings and multiple iterative discussions between the analysts. Themes were reviewed in relation to the whole dataset and refined with a clear definition through further engagement with the transcripts. The collaboration between the two analysts enabled a richer discussion of the complexities within participant narratives and supported the credibility of the analysis.

## Results

Four themes incorporating eleven subthemes were identified, as mapped in Fig 1. The first three themes explore individual activities used to support perinatal mental health, sociocultural pressures and barriers to care, with the final theme encapsulating how the experiences and preferences of participants could inform future culturally inclusive music-based provision. Each theme is discussed below with indicative quotations from the focus group and interview transcripts. Additional excerpts from the transcripts can be found in S1 Table.

### 1. Supportive mechanisms during the perinatal period

**1.1. Emotional and practical support from family and friends.** Participants valued help from their family and friends during their perinatal experience, particularly for practical support during mental and physical recovery post-birth, but also for feeling emotionally uplifted. While some women felt comfortable reaching out to family and friends for support, others appreciated family members who took the initiative to provide assistance without them needing to ask for help.

"I was so grateful that my mum made me move into her house after I came back home from the hospital, because I don't know how I would have coped. She was literally the one that forced food down my mouth." (Focus Group 1)

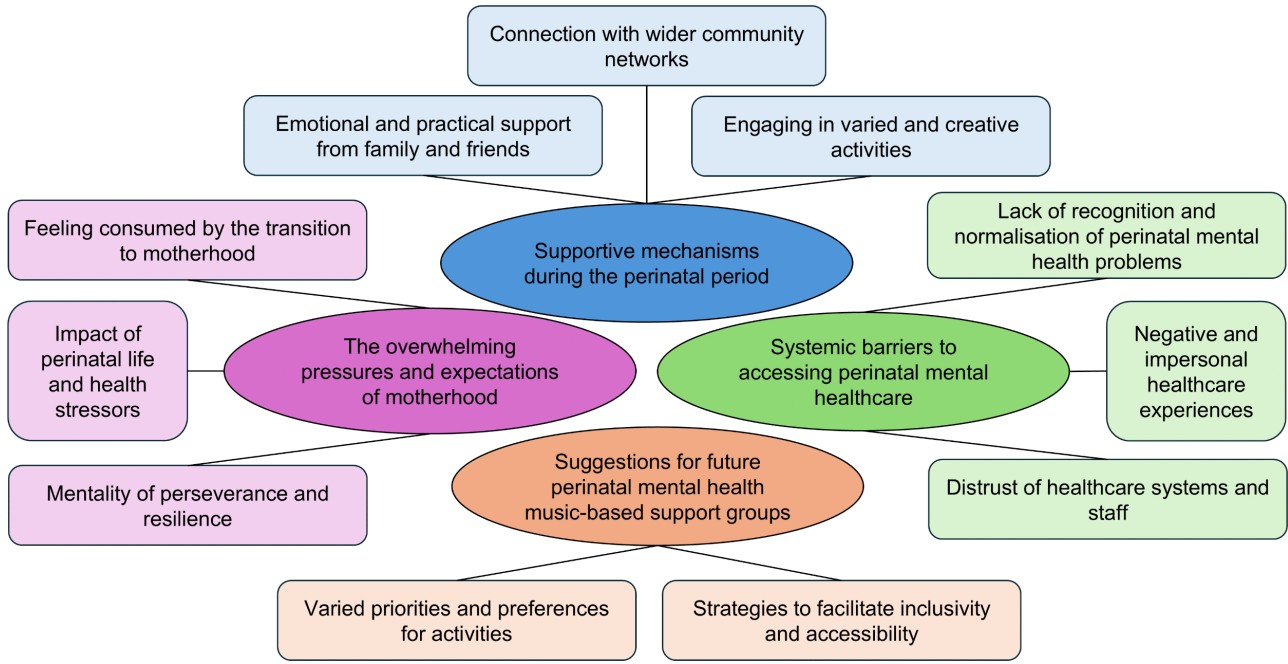

**Fig 1. Themes and subthemes from the reflexive thematic analysis.**

While many relied on family and friends, several participants expressed challenges with these relationships during the perinatal period, as family members or friends could become critical or domineering. In particular, family disagreements related to cultural expectations and approaches to parenting created additional difficulties for women navigating motherhood.

"There's no understanding there all the time, or you don't feel you get as much support because they're not in your position. Or you're judged for being in that position" (Focus Group 4)

For some participants, having a supportive partner was most important, emphasising the "burden" and "struggle" for those without a partner (Focus Group 2). Additionally, logistical barriers to familial and friend support networks, such as social distancing during the COVID-19 pandemic or family living in a different country, left some participants feeling isolated during the perinatal period. This emphasises the importance of developing peer support networks between women, as an alternative or additional outlet to connections with family and friends.

**1.2. Connection with wider community networks.** Where participants had limited support from family and friends, wider social networks in the community were especially important for sharing experiences and feeling less isolated.

"I would say meeting up with other mothers… it's a big thing... it's sharing your journeys together and realising that somebody is going through the same thing as you" (Interview 3)

The opportunity to meet other parents with similar experiences or struggles of their own facilitated feelings of normalisation and mutual understanding for participants. Community groups could also provide a sense of perspective and grounding for women, encouraging them to be more compassionate towards themselves, and empower others during their journey through motherhood by sharing advice. However, some participants spoke about how these community

groups made them feel excluded, partly due to a lack of representation of participants from similar cultural or socioeconomic backgrounds. This made some participants feel unable to share their experiences, worrying about judgement from other parents.

"I always feel like there needs to be a Black NCT [National Childbirth Trust] group … where I could meet like-minded mums that have some shared experiences." (Focus Group 2)

Participants also discussed the lack of available community support, which left them feeling even more isolated during the perinatal period, exacerbating mental health problems.

"the lack of support is actually causing a lot more issues where women are feeling more and more isolated, feeling more and more alone and despondent." (Interview 2)

Participants may have struggled to find safe spaces to share their experience due to a lack of appropriate provision or a lack of awareness of available support options. Future culturally inclusive interventions should consider the importance of connecting women with others they can relate to, such as specific groups for Global Majority women, and exploring effective advertising strategies to increase the visibility of community support.

**1.3. Engaging in varied and creative activities.** Participants discussed a wide range of activities used for support during the perinatal period, such as walking, journalling, reading and shopping. Creative activities, including listening to music and singing, were often mentioned and associated with a broad range of benefits, such as improved mood, empowerment and emotional release.

"when we listen to songs and the words that are in the songs, there's a healing behind it, there's emotion behind it. And it kind of allows you… it forces you to recognise that if you need to cry, you cry." (Focus Group 1)

Music listening also enabled mothers to feel more present and relaxed, offering a mechanism for emotional regulation during times of significant stress.

"I loved listening to music. Music was always my kind of outreach for just relaxing or just sitting in the park and grounding my feet on the grass when the weather was good." (Focus Group 4)

Others described the impact of dancing to music to help them feel empowered and connected with their infant. The idea of music and dance bringing people together was highlighted several times throughout the focus groups and interviews as a way to transcend language and communication barriers. However, several participants did not use music-based activities during the perinatal period and each had a specific preference for the type of supportive activities that brought them comfort. As the below quotation demonstrates, lack of motivation and poor physical or mental health could reduce women's ability to engage in group activities.

"I didn't want to talk to people or be around people and so I think not having the pressures of interaction with others, that's like organised fun, will mean that you can have some type of outlet in your own setting." (Focus Group 3)

## 2. The overwhelming pressures and expectations of motherhood

**2.1. Feeling consumed by the transition to motherhood.** Participants often described feeling that their social identity as a mother had become all-consuming, emphasising the overwhelming changes to routine and prioritisation of

their infant's needs over their own. Women experienced significant changes across all aspects of their life when becoming a mother, in addition to existing responsibilities regarding work and household duties.

"it's fine to grieve your old life, because it's a massive change to have nobody dependent on you and just being able to get up and go about your daily life to suddenly having someone solely dependent on you." (Interview 4)

The loss of personal identity, independence and time to consider themselves was challenging for most participants, highlighting the importance of groups offering space for down-time and self-care separate from motherhood. For some participants, the uncertainties of pregnancy, birth and motherhood were also overwhelming.

"I don't think we're good at being okay with not knowing. I think the fear of that is really crippling a lot of people sadly in this and it's causing a lot of anxiety" (Interview 2)

Participants described the potential harm of comparing their experiences with unrealistic societal expectations of motherhood, especially when participating in community groups. Separating expectation from reality enabled some participants to better support their perinatal mental health.

**2.2. Mentality of perseverance and resilience.** Several prominent sociocultural expectations regarding the perinatal period were highlighted as key barriers to accessing support. One prominent expectation was that women should persevere through their difficulties and show mental resilience. Some described this in a positive light, discussing ways to work on their mental health productively, remove negative thought patterns and develop coping strategies. However, most participants feared judgement and scrutiny from others if they did not live up to the expectation of resilience and perseverance.

"Especially culturally, a lot of women of colour or ethnic minorities are expected to just get on with it and it's a blessing you had a child, you shouldn't be miserable – 'the baby blues' as some people call it. In some cultures, they don't even call it that, they just tell you to get on with it." (Interview 1)

This expectation of resilience has likely developed through several interrelated factors, including cultural understandings of motherhood, expectations from society, lack of awareness of perinatal struggles, lack of appropriate support avenues and distrust of health services. Additionally, women may believe that admitting their struggles is a sign of weakness and failure. Perhaps due to these expectations, exacerbated by cultural perceptions and traditions, one participant felt that women of African, Caribbean and Mixed heritages were generally not accessing community groups or other avenues of support for their mental health.

"I guess for me as well, culturally, Black women don't go to these kinds of groups essentially. You know, you kind of just stay at home, get on with it" (Focus Group 1)

**2.3. Impact of perinatal life and health stressors.** This subtheme relates to the various life pressures faced by the participants and how these impacted stress levels and overall perinatal mental health. Financial stress was a major contributor to poor perinatal mental health, especially worries around keeping a job and being able to financially support their new family.

"you might have a low mood because of the actual hormonal impacts, but then … you're also thinking about actually the financial implications of having a child." (Focus Group 3)

Participants also discussed the impact that poor physical health had on their mental health and ability to bond with their infant. Some described the lack of support and understanding for debilitating physical perinatal symptoms, as well as the challenges of high-risk pregnancies and traumatic births.

"I spent a huge majority of my time just in a dark room in bed, vomiting, curtains closed and not being able to eat or function. And that had a huge effect on my mental health" (Focus Group 1)

Participants struggled with the fact that these stressors could be unpredictable, unavoidable, debilitating and could significantly impact their mental health. While these stressors are likely to be commonly experienced among women from different backgrounds, they could be exacerbated by several demographic factors, such as being working class or a single parent. This subtheme suggests the importance of carefully considering the logistics of future interventions to avoid burdening women with additional stressors related to participation.

### 3. Systemic barriers to accessing perinatal mental healthcare

**3.1. Lack of recognition and normalisation of perinatal mental health problems.** Despite their prevalence and potential severity, participants felt that perinatal mental health problems were not regularly discussed in their communities, including among family, friends and healthcare professionals. The lack of discussion around perinatal mental health could perpetuate societal stigma and reduce the likelihood of women seeking support due to fear of being judged or perceived as incapable.

"Just knowing that it's not a phase or a facade, so be more advertised widely or talked about more widely by the practitioners, the GPs, the midwives, the consultants as a normal thing, rather than it be a cliche thing" (Interview 1)

Participants also reflected on how healthcare professionals could be dismissive of perinatal mental health problems, typically directing their attention towards physical health checks and the health of the infant. Additionally, family could underplay the severity of poor perinatal mental health, especially in certain cultural contexts with less awareness of perinatal mental health problems. In part due to this lack of recognition, several participants did not realise they had mental health problems and were not aware of how to access support.

"I didn't even realise it was postnatal depression. Again, despite my profession [as a midwife], I didn't recognise it" (Focus Group 1)

Although awareness of perinatal mental health problems has improved over time, this subtheme highlights that there is still significantly more work to be done to reduce this barrier to accessing perinatal mental health support and improve the rates of diagnosis and treatment. Psychosocial and community interventions could effectively mitigate barriers related to stigma. Furthermore, by incorporating elements of peer support and psychoeducation, they could help to validate women's mental health challenges and encourage future help-seeking.

**3.2. Negative and impersonal healthcare experiences.** Participants described various experiences with healthcare services where their preferences were disregarded and person-centred care was rarely evident across participants' perinatal experiences. Appointments were not sufficiently tailored to best support individual needs and often became a tick box exercise.

"I gave them my birthing plan and the midwife or the consultant that was on that night … they didn't read it. Everything I did not want, they did." (Focus Group 4)

"it's very clear they want to get you out of the door as soon as possible." (Interview 2)

These impersonal healthcare experiences led women to feel undervalued and decreased the likelihood of them sharing potential concerns, especially those that may be sensitive in nature. Although not explicitly discussed in the focus groups

and interviews, participants' impersonal healthcare experiences could also imply a lack of cultural sensitivity. Women of African, Caribbean and Mixed heritages may be less likely to receive care from a health professional from a similar background, which could increase their likelihood of feeling alienated from services.

Additionally, the use of negative language by healthcare professionals sometimes worsened experiences of poor mental health when women were made to feel incompetent or lesser. This was exacerbated by a lack of continuity and regularity of care, which reduced the number of opportunities for sharing problems and increased the amount women had to repeat their concerns. Systemic issues with the healthcare service, such as overstretched staff and lack of training in perinatal mental health, may be key contributors to the variability in quality of care. With healthcare services struggling to meet their aims of providing person-centred care, community-based interventions may have the greatest capacity to engage with individuals and their communities to co-create care that addresses their specific health needs.

**3.3. Distrust of healthcare systems and staff.** Many participants emphasised hesitations around disclosing perinatal mental health problems due to fear of repercussions on their infant or other children. This lack of trust could also be exacerbated by previous negative healthcare experiences.

"there's no way I'm gonna talk to my GP or health adviser and say I'm feeling depressed. They're gonna take my baby away. They're gonna think I'm not capable" (Focus Group 3)

One participant discussed how women of African, Caribbean and Mixed heritages may be more likely to experience judgement from healthcare professionals as being not fit to look after their children.

"We should be able to talk about what we need to talk about, same as a White person should be able to talk without judgement, without any preconception, misguidance, whatever word you want to use. But as you know, we're living in a White society. There is this judgement for Black women" (Focus Group 2)

However, some women found greater trust in healthcare professionals when compared to family, as healthcare professionals have more specialist knowledge about perinatal mental health problems and the different support options available.

"I find it easier to speak to somebody who's not family than to speak to family. Family will … not actually take your concerns into consideration. Whereas somebody who's trained within that field, I would hope, would be able to talk to me about what my options are." (Focus Group 4)

The idea of trust could be a key foundation for developing future culturally inclusive interventions, with women being more likely to approach someone who they have had positive previous experiences with, has been endorsed by others in their community, understands them and their background and gives them sufficient space to voice their concerns. Interventions could consider ensuring that professionals represent similar backgrounds to participants, although, in some cases, this could increase feelings of expectation and judgement.

## 4. Suggestions for future perinatal mental health music-based support groups

**4.1. Varied priorities and preferences for activities.** Participants had diverse preferences around the use of music in perinatal mental health support groups and the types of activities that would be most engaging. Some participants expressed significant interest in group singing during the perinatal period, depending on the type of musical activity.

"Every time I've done karaoke with people that I don't know, I've had a great time. … So I think that kind of singing is great" (Focus Group 3)

Other participants expressed hesitation towards group singing, including concerns about feeling self-conscious, lacking expertise and feeling intimidated in a group setting. Whilst acknowledging that music-based groups are not for everyone, future interventions could consider different strategies to negate potential anxieties about participation, such as emphasising the collective nature of the activity or ensuring it is easy to join in.

"I'm giving myself an extra thing to be anxious and worried about, so … I think for me it wouldn't – as much as I love music and I know lots of songs – it would be a turn off for me." (Focus Group 2)

Karaoke was suggested several times to accommodate different genre preferences and engage participants in an enjoyable musical activity. Others identified more with songwriting as a mechanism to reflect on personal experiences and create memorable songs to use with their infant. This could be an especially useful support mechanism for women who are struggling to talk about their experiences with others, as well as enabling them to connect with aspects of personal and cultural identity. While many of the suggested activities offered nurturing components for both the women and their infants, some participants emphasised the importance of prioritising women's needs and offering a space for respite from the potential challenges of motherhood.

"I was kind of thinking, probably selfishly, but I've always thought it was more me-time, this is a time for me to go out and do something for me, like on my own." (Focus Group 4)

Overall, this sub-theme emphasises the importance of considering variety and choice, both within and across perinatal interventions.

**4.2. Strategies to facilitate inclusivity and accessibility.** This subtheme explores how perinatal support groups can utilise a flexible and diverse approach, prioritising each individual's needs to increase inclusivity. Participants typically preferred groups to be scheduled in the middle of the day to best accommodate conflicting responsibilities, such as childcare for older children, and spread out across a range of locations. Participants also discussed the benefits and drawbacks of conducting perinatal support groups online versus in-person. While the online format was logistically easier and presented less physical barriers to accessing the group, participants emphasised the power of having in-person sessions for a more connected and engaging experience.

"I like the idea of online. So you've not got the stress and the worry of organising your child and getting them ready. But then also, I do like in-person as well. So a choice of both?" (Interview 1)

These logistical factors are particularly pertinent for culturally inclusive interventions, where women may be less likely to have nearby family to offer support for childcare. Furthermore, cultural inclusivity was also a key consideration to enable participants to identify themselves in both the musical content and group demographics. For example, culturally inclusive music-based interventions for perinatal mental health could foster a space for women from similar backgrounds to feel acknowledged, but with some diversity of cultures to enable opportunities for cross-cultural learning. Participants discussed the importance of incorporating a variety of musical genres, including popular music from different decades and multicultural songs from around the world.

"Are we cognisant of other cultures? Are we having African music, for example, and Afro Caribbean music and other cultural music? Or is it just going to be Mozart? Or a certain type of music, because even in those little nuances, we can unpick that and say, okay, this might not be for me." (Interview 2)

Several other strategies for accessibility were suggested, including subsidised costs for those with lower socioeconomic status and accommodating the needs of neurodivergent participants.

## Discussion

Our findings encompass the varied lived experiences of fourteen women of African, Caribbean and Mixed heritages living in South East London. Participants outlined a wide range of individual, sociocultural and systemic barriers faced during the perinatal period, the activities they used to support themselves and suggestions for future music-based interventions for diverse women. Many of the systemic issues with perinatal mental healthcare evident across participants' experiences in this study appear common to women from other backgrounds and contexts. Similarly to our findings, other research has suggested that barriers to perinatal mental healthcare across the UK include being scared of social services, fear of judgement, lack of social support, limited awareness and previous negative experiences of healthcare [10]. However, culturally-specific expectations, experiences of stigma and mistrust played a significant role in heightening the negative impacts of various barriers to perinatal care. This could explain the prevailing perinatal health inequalities associated with Global Majority ethnicities [12,13]. These experiences are further explored below, with an emphasis on how they can offer specific suggestions for future culturally inclusive music-based interventions for perinatal mental health.

Participants often used informal support mechanisms to maintain and improve their perinatal mental health. Self-care behaviours, such as physical exercise and relaxation techniques, have been associated with improved perinatal wellbeing [49,50]. Participants discussed how informal sources of support often replaced the use of formal services, which aligns with the low rates of help-seeking and diagnosis of perinatal mental health problems in prior research [3,11]. This reliance on informal support could be partially explained by the pressures of mental resilience and social stigma that participants faced [21], deterring them from seeking mental health support [51]. This explanation is supported by a recent systematic review, which found that Black Caribbean and South Asian women described depression as culturally unacceptable and reported an inherited cultural legacy of resilience and strength [52]. These preferences for informal support could also be a result of the systemic ethnicity-related inequalities and discrimination embedded in statutory mental healthcare services, as identified in several previous studies [17,27]. Improving the provision of informal and community-level support may help to mitigate some of the significant barriers that women of African, Caribbean and Mixed heritages experience related to stigma and societal prejudice [21,23].

Participants also emphasised the importance of strong social connections with partners, family, friends and other community members, which aligns with prior research on the benefits of partner and peer support in the perinatal period [27,53]. However, several women discussed the lack of available avenues of community support when family were not living nearby, especially during the COVID-19 pandemic, which reflects broader research on the social barriers to perinatal mental healthcare [10]. Community connection was highlighted by participants as a key mechanism to incorporate in future interventions, as social isolation remains highly prevalent in the perinatal period and has a central role in experiences of perinatal depression [54]. Indeed, music-based interventions could align well with this need, due to their ability to foster a sense of belonging, group connection and social support [30]. However, participants also indicated the importance of a nuanced approach, acknowledging how women can feel excluded from community groups due to a lack of ethnic diversity and difficulties connecting with those from different backgrounds. Prior research confirms the prevalence of ethnic and other socioeconomic inequalities within community perinatal mental health interventions in the UK [28,55]. Providing culturally aware and person-centred approaches might encourage more engagement from women of African, Caribbean and Mixed heritages and tackle broader socioeconomic inequalities embedded in community interventions [22,29,56].

This study identified a range of personal barriers to seeking mental health support, including financial pressures, difficult relationships and stresses around physical health. However, many of the significant barriers experienced by participants were embedded within existing healthcare provision or wider sociocultural norms. Participants discussed the limited awareness and recognition of perinatal mental health problems amongst their families, friends and sometimes healthcare providers, which aligns with similar prior research in other UK settings [10,56]. Specifically, research has highlighted how some people from Global Majority backgrounds are either not aware of perinatal mental health problems, do not consider

their symptoms to be an illness, do not have the language to describe their symptoms as a disorder or adopt alternative explanations for their symptoms [52]. This barrier may be exacerbated by cross-cultural differences in language, with some cultures lacking a direct translation for depression or using terminology that underplays the potential severity of symptoms. For example, the term "baby blues" was used to describe postnatal depression, despite its definition referring to a shorter-term change in psychological state due to hormonal fluctuations after birth [57]. Participants also discussed fear of repercussions and negative experiences with healthcare professionals as a barrier to seeking help. Stigma and mistrust of healthcare services are widely recognised barriers to healthcare for diverse populations [58,59]. Promoting culturally sensitive discussion around perinatal mental health across society is necessary to address different sociocultural understandings of mental health and provide diverse populations of women with support strategies. To fully address these barriers experienced by African, Caribbean and Mixed heritage women, future interventions may need to move beyond ideas of cultural competence to develop equitable and interdependent partnerships with diverse cultural communities, including those with more traditional approaches to healing [60].

Although a wide variety of perinatal coping strategies were discussed, creative activities were often mentioned as a useful form of support. Women discussed several benefits of engaging in music for perinatal mental health, such as emotional expression, healing, relaxation, empowerment and connection, which are comparable to mechanisms identified in prior research [36]. Our study demonstrates the significant potential benefits of developing more participatory music interventions for perinatal mental health within different community settings. It is especially important to co-develop these interventions with women of African, Caribbean and Mixed heritages, who may be deterred from engaging in perinatal mental health music-based groups due to a lack of inclusivity and diversity. This will address the current paucity of perinatal music research on culturally inclusive interventions and align with recent UK policy recommendations on improving equitable access to musical care during the beginning of life [61].

This study also extends prior research by exploring different preferences regarding musical support from the perspectives of women with lived experience. Some women discussed their preference for a certain type of musical activity, such as songwriting without singing or participant-led karaoke instead of structured sessions, and a minority of participants did not wish to use music to support them perinatally. The potential unintended consequences and barriers to engaging in music-based interventions have not always been considered in the field of arts and health [62], but this study highlights the importance of considering these nuances and diverse preferences during music-based intervention development.

### Recommendations for future music-based interventions

Overall, our analysis indicates three broad areas of consideration for future participatory music-based interventions in community contexts to ensure inclusivity of women from diverse ethnic backgrounds: (1) sociocultural awareness, (2) logistical factors and accessibility and (3) musical considerations (see Fig 2). These recommendations nuance existing research on the components, goals, mechanisms and outcomes of music interventions for perinatal mental health in varying community and clinical settings [63]. Across these recommendations is an overarching consideration of inclusive practices, including community-specific understandings of social stigma, prejudice, cultural pressures and expectations that may be experienced by intervention participants. While providing some specific suggestions to guide future interventions, these recommendations emphasise the importance of different options to support diverse preferences and modes of engagement. In addition to intervention development, this research could also guide inclusive approaches to the implementation of future participatory music-based interventions.

### Limitations

One limitation of this study is the positionality of the two analysts as White researchers seeking to research the differentiated experiences of women of Global Majority ethnicities, where a lack of understanding of their positioning and subjectivities could lead to misrepresentation [64]. We incorporated several approaches in our analysis to mitigate this potential

**Fig 2. Sociocultural, musical and logistical recommendations for an inclusive perinatal mental health music-based intervention.**

limitation, including meaningfully centralising the voices of participants through ongoing reference to the transcripts and reflexive journaling. We also incorporated regular discussion with the local community leader and expert by experience (TA), which ensured the developing themes fully reflected the personal experiences of women of African, Caribbean and Mixed heritages. TA led the interviews and focus groups to enable deeper connection with participants and a richer exploration of personal experiences. Additionally, we acknowledge that some individuals may not resonate with the terminology used to describe ethnicity in this study and future research is warranted to explore the language that best reflects the preferences of Global Majority individuals.

Another limitation of this study is that the focus groups and interviews were completed online and in English, which may have excluded participants with limited digital literacy or knowledge of the English language. The online nature of the focus groups and interviews may have limited participant engagement, as rapport was more difficult to establish and instances of poor internet connection affected the depth of discussion. The scope of this paper was limited to South East London, so the findings may not be representative of other contexts across London or the UK. Some of the themes from this study have been found in other communities around the world who experience ethnicity-based inequalities in perinatal mental health [65], but future community-specific inquiries should be made in other contexts to ensure interventions meet the needs of their local population.

## Conclusion

This qualitative research study demonstrates some of the key narratives prevalent across the perinatal experiences of women of African, Caribbean and Mixed heritages, including (1) supportive mechanisms during the perinatal period, (2) the overwhelming pressures and expectations of motherhood, (3) systemic barriers to accessing perinatal mental healthcare and (4) suggestions for future perinatal mental health music-based support groups. This paper provides several

important considerations to enhance the cultural sensitivity of future perinatal mental health community interventions. Additionally, this study highlights how coproduction of perinatal mental health interventions with women of African, Caribbean and Mixed heritages is essential to address implicit health inequalities. Future research is warranted to explore how the recommendations elicited from this study can be realised through a culturally informed participatory music intervention for perinatal mental health.

## Supporting information

**S1 Table. Additional supporting excerpts for each theme from the focus group and interview transcripts.** (DOCX)

## Acknowledgments

The authors would like to thank everyone who participated in an interview or focus group.

## Author contributions

**Conceptualization:** Katie Rose M. Sanfilippo, Malik B. Jeng, Lauren Stewart.

**Data curation:** Juliet Firth.

**Formal analysis:** Lottie Anstee, Juliet Firth.

**Funding acquisition:** Malik B. Jeng, Lauren Stewart.

**Investigation:** Toyin Adeyinka.

**Methodology:** Lauren Stewart.

**Project administration:** Juliet Firth, Toyin Adeyinka, Malik B. Jeng.

**Supervision:** Katie Rose M. Sanfilippo, Lauren Stewart.

**Writing – original draft:** Lottie Anstee, Juliet Firth.

**Writing – review & editing:** Lottie Anstee, Juliet Firth, Toyin Adeyinka, Katie Rose M. Sanfilippo, Malik B. Jeng, Lauren Stewart.

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
