## [Decision Letter · Decision Letter 0]

9 Dec 2025

PMEN-D-25-00439

Exploring the experiences of women of African, Caribbean and Mixed heritages to inform a music-based intervention for perinatal mental health in South East London: a qualitative study

PLOS Mental Health

Dear Dr. Anstee,

Thank you for submitting your manuscript to PLOS Mental Health. After careful consideration, we feel that it has merit but does not fully meet PLOS Mental Health’s publication criteria as it currently stands. Therefore, we invite you to submit a revised version of the manuscript that addresses the points raised during the review process.

We look forward to receiving your revised manuscript.

Kind regards,

Gellan Karamallah Ramadan Ahmed

Academic Editor

PLOS Mental Health

Journal Requirements:

1. In the online submission form, you indicated that The qualitative data that support the findings of this study are available on reasonable request from the corresponding author. The transcripts are not publicly available to protect the privacy and confidentiality of participants, as they contain potentially identifiable and sensitive information.

3. Uploaded as supplementary information.

2.  Please insert an Ethics Statement at the beginning of your Methods section, under a subheading 'Ethics Statement'. It must include:

1) The name(s) of the Institutional Review Board(s) or Ethics Committee(s)

2) The approval number(s), or a statement that approval was granted by the named board(s)

3) (for human participants/donors) - A statement that formal consent was obtained (must state whether verbal/written) OR the reason consent was not obtained (e.g. anonymity). NOTE: If child participants, the statement must declare that formal consent was obtained from the parent/guardian.

Reviewers' comments:

Reviewer's Responses to Questions

**Comments to the Author**

1. Does this manuscript meet PLOS Mental Health’s publication criteria?

Reviewer #1: Partly

Reviewer #2: Yes

2. Has the statistical analysis been performed appropriately and rigorously?

Reviewer #1: N/A

Reviewer #2: No

3. Have the authors made all data underlying the findings in their manuscript fully available (please refer to the Data Availability Statement at the start of the manuscript PDF file)?

Reviewer #1: Yes

Reviewer #2: Yes

4. Is the manuscript presented in an intelligible fashion and written in standard English?

Reviewer #1: Yes

Reviewer #2: Yes

Reviewer #1: Title: Exploring the experiences of women of African, Caribbean and Mixed heritages to inform a music-based intervention for perinatal mental health in South East London: a qualitative study

1. Comments to Author

1.1 Overview and general recommendation

*The following paper describes a qualitative study focusing on focus groups and interviews with 14 women of African, Caribbean and Mixed heritage to explore their perinatal experiences and preferences regarding music-based support to inform a culturally inclusive perinatal participatory music intervention. It describes four overarching themes and provides some future recommendations and directions for research.

Thank you so much for sharing your work and your manuscript. It has been a pleasure to read and review this paper. This paper meets the scope and partly meets the criteria for publication as it stands. There is a strong argument made throughout, and this is work that will have value for future culturally inclusive music interventions.

However, there are a couple of areas where I encourage you to further develop your ideas to take this work to the next level. Particularly, I would encourage you to look back at your methodology and results when it comes to your position statement of taking a critical realist stance. At the moment, it is unclear how the critical realist philosophy has been woven into the work and themes, which creates a misalignment. You will see many of my comments are about what you say you have done vs the critical realist stance you say you have taken. You could do constructivist or critical realist for this work, both appear appropriate. But whatever you choose, it must be consistent throughout. There is also a possibility to move beyond description in your results and engage a little further with the data. Please see specific comments below that hopefully should be able to help and guide the revision, all in the good spirits to advance this already great piece of work.

2 Major comments

2.1 Title and abstract

*In the title, you rightly mention that it is a qualitative study. Is there a reason why you don’t mention that this is a critical realist qualitative study? I am just a little confused as to the stated philosophical position and the lack of engagement with that philosophical position throughout. If the intention is to go through a constructivist lens (which currently reads more as such), then that is absolutely fine, but I wonder how much critical realist you are adding or not adding.

* You mention in the abstract (line 24) that inequalities occur at different levels of the system, which necessitate a greater reliance on self-coping strategies. It might be helpful to clarify in the introduction how self-coping strategies operate differently across these levels, or why they may be more effective at some levels than others, to guide the reader and position the music intervention within one/various layers of the system you mention.

*In line 35, you highlight the individual experiences of the perinatal period, which is absolutely fine, but maybe through a critical realist stance, you may want to share the dynamics between structure and agency and some of the repercussions that have on creating culturally inclusive interventions.

2.2 Introduction

*From lines 59 to 64, you mention some great information that from a critical realist stance can be very useful (i.e. systemic issues, looking at the problem from a more emancipatory lens, critical analysis of power relationships and value-laden aspects of science…). However, at the moment, it is placed as background information. If you are going to work on the critical realist stance, make sure that you align your wording and examples to reach philosophical coherence.

*There is a good discussion from lines 79-91 around community-based services, which is well referenced and justified. However, it then moves to interventions, which are slightly different to services. I.e. building culturally inclusive services and interventions. I think the confusion lies here that you are interchangeably using in these lines the words services, interventions and programmes. It would be good to keep consistent in language to not confuse the reader as to whether you are talking about services or interventions, and make sure to build the gap as to the need to work on both.

*For your research questions and text beforehand, it reads very much from a constructivist lens. Again, no problem if that is your intention, but there is a philosophical misalignment here between the needs of a critical realist approach, which is beyond experiences to the powers, underlying mechanisms, structure vs agency and other concepts in comparison to more constructivist experiences.

2.3 – Methods

*Is there a rationale as to why some participants did focus groups and some others did interviews? What are the potential implications of this decision? As well as the numbers of each, having four focus groups of smaller numbers rather than two or one of larger numbers?

*You may want to add a reference or justification as to why your sampling techniques were the most appropriate to meet your study aims.

*Could you elaborate on the rationale for using the same questions in both the focus groups and the interviews? Given the distinct purposes and interactional dynamics of these methods, it would be helpful to understand how this decision supported your study design.

*Is there a reason why the questions used did not directly address the cultural relevance of the interventions? As some of the questions are broader for “women” (such as question 6), was there clarification that this was to build culturally relevant interventions as well?

*The biggest concern here is lines 197-201 and the description of critical realism as the philosophical stance. I would encourage you to use some references here, explain to the reader what critical realism means and what the implications are of adopting such a lens for your study beyond the description you currently provide. For instance, there is no mention of depth ontology, ontological realism, power dynamics, etc. There is a slight comment on agency vs structure, but it is not substantiated enough, i.e. could add Archer's morphogenesis, for example, here. It is a philosophical basis rather than a theoretical one. This is not a methodological paper, so I wouldn’t expect to see a full description, but it is an important claim made that then impacts the analysis, and, as mentioned previously, I don’t see how the critical realist stance informed your results for now, as they read more constructivist. For instance, there is no mention of mechanisms, retroduction or other critical realist language I would expect to see.

2.4 – Results

*My main observation in the result section is twofold, and this is relevant for all sections and themes, hence it all comes in one paragraph. Firstly, there is a lot of great descriptive content that shares ideas around the themes, but it remains superficial at times and does not engage deeply with the depth ontology or mechanisms, which, if using a critical realist stance, I would expect to see. For example, in 1.1, there is a description of participants' value on family and friends, others expressed challenges, and others felt isolated. The content you could consider here (and throughout the other sections) is – so what? What is it about having critical relationships or being domineering that helps us understand the needs of the culturally inclusive music interventions better? In the discussion, you go into more detail, but it would be good to expand the results with more interpretation and analysis, rather than description. The second element in the results is (sorry for sounding repetitive) the critical realist stance. I don’t see the philosophical alignment. You mention relationships between individuals and contexts more widely, but there is a lack of a narrative that aligns with the aims of critical realism. I encourage you to look at this section and 1. Increase the analytical depth and 2. Align more with the critical realist stance you took to showcase how this philosophy of science impacted your analysis and results. There is good work by Wiltshire, Ronkainen or Fryer on how to do Realist Thematic Analysis. I see this is not what you did here as described by these authors, so there is no expectation to do the same, but it may give you some ideas on some of the wording or things to include or indeed consider the philosophical stance.

2.5 - Discussion & Conclusion

*This section may change slightly based on the thoughts above on how you decide to weave (or not, if going for an interpretivist lens) the critical realist lens. For example, in the first paragraph, do you have any thoughts on structure vs agency on that particular point of systemic ethnicity-related inequalities in mental health care? There is a great book coming soon on this (Decolonizing Global Health: A Critical Realist Perspective).

*In your limitations section – look at the comment below in minor comments around language. I would also encourage you to move away from saying that your study is non-generalisable. That is not a limitation of a qualitative study, after all, it has different rules and assumptions about generalisability than quantitative work, and the aim is not to generalise. But you may want to talk about transferability (or theoretical generalisability instead of statistical generalisability) instead, as a concept, which appears more appropriate for this research.

3. Minor comments

3.1 Title and abstract

*In line 30, you can be more specific as a perinatal mental health participatory music intervention, as then you differentiate between perinatal care more widely and perinatal mental health. You can do the same in line 34, suggestions for future perinatal mental health creative support groups. Have a look throughout the manuscript to see where you could be specific about perinatal vs perinatal mental health care. I have seen a few examples throughout.

*There is again no mention of the critical realist position taken in line 31, described as an inductive reflexive thematic analysis. You may want to add informed by a critical realist stance or philosophy, and then explain more later on how that has impacted your interpretation of results.

3.2 Introduction

*Line 51, you may want to spell out a couple of what those healthcare barriers may be.

*Lines 75-78, a very good example of the WHO's need to move towards community-based. Any UK-based examples you want to share, i.e. NHS 10-year plan or similar, to position the need from a UK perspective too?

*You use the word mechanism in line 93. If you are using a critical realist stance, be careful with the wording of the mechanism and how it will be defined, as in critical realism, these have a very specific meaning.

*In lines 107-108, you highlight “little consideration about how musical activities can be integrated into inclusive community groups”. This is an implementation question, not an intervention question. Your aim in the abstract is “inform a future culturally inclusive perinatal participatory music intervention”, not necessarily about how to integrate them. We need to know what they are first, their components and their effectiveness before we can fully integrate them. Therefore, you may want to consider rewording this sentence to focus on the intervention, rather than the implementation, to meet the needs of the study.

3.3 – Methods

*You use fourteen in line 145 but 14 in the abstract. Remain consistent as to which one you are using.

*You mention in the inclusion criteria in line 146, “women who have given birth to at least one infant”. Does this mean that the women interviewed or included could have had an older child and have already gone through the perinatal period? Just to clarify for readers to understand if women could have been experiencing the perinatal period at the time of the interview, or retrospectively share their experiences.

*In Table 1, it could be useful to highlight or share how each question meets the five topics shared in the preceding paragraph, maybe at the end of each sentence with a (1) or similar, to showcase how each question aligns with the topics.

3.4 – Results

*Figure 1 – You may want to make the distinction between the different themes and subthemes clearer, maybe through colour coding each theme and subtheme or through labels?

*Is there a reason you mention creative support groups rather than music-specific ones? That appears as more general rather than specific to the music needs of the intervention you are trying to co-develop. Of course, music is creative, but is it under that umbrella, and it shows that you want to tackle the music specifically.

*There is a lot of information on barriers and service use (or lack thereof) for the participants, which is important contextual information. However, I see that the cultural elements are less emphasised throughout, except in a couple of elements. As this is a specific study looking at culturally inclusive music-based interventions, I wonder which of the results you suggest are either specific to cultural inclusivity or wider problems with healthcare services and dynamics. It may be worth considering this to really engage with the question.

3.5 - Discussion & Conclusion

*Line 447 – if using word mechanisms and a critical realist stance, be careful in how mechanisms are worded and conceptualised throughout. Mechanisms, as per Mingers and Standing (2017) through a critical realist lens, are “a structure of inter-related parts together with the powers or tendencies that the structure possesses”. Or can see Bhaskar's work for his conceptualisation. The point is to be clear about what definition you are using, so that there is alignment in what you mean.

*The strong social connections paragraph is well developed, but is there value in positioning this within the music intervention per se too?

*The conversation on language and stigma is really interesting again from a critical realist stance in lines 480-489. I am not sure if you have already, but Braun and Clark, in their recent book, have a section on this, citing the work of Maxwell. You may find it interesting to develop these ideas further.

*Fig 2. May want to make the title specific to music interventions rather than community perinatal mental health support interventions, as there could be many! You may want to comment in the recommendations for future music-based interventions about implementation science and how this research could also be used to help move us from not creating an intervention only, but also to achieve health equity through implementation science approaches.

I hope this review is helpful and clear, all in the spirit of advancing scientific knowledge and supporting you in taking this important work to the next stage. Thank you for your efforts!

Reviewer #2: The Introduction should conclude with an explicit problem statement, as the current formulation remains implicit and may leave readers unclear about the study’s precise focus. Additionally, several parts of the Introduction reiterate similar ideas about healthcare inequities and mistrust; consolidating these points into a more streamlined paragraph would improve readability and coherence. In the Results section, some subthemes rely heavily on lengthy participant quotations. Shortening these excerpts and expanding the accompanying analytical narrative would help achieve a stronger balance between interpretation and evidence. The Discussion would also be strengthened by more consistently linking each theme to existing research, ensuring that the findings are situated within broader scholarly conversations. Finally, the coding process needs more detailed description. Specifying the steps taken (such as line-by-line coding, independent coding by multiple analysts, and iterative refinement of codes and themes) will allow readers to fully understand and assess the robustness of the thematic analysis.

**Do you want your identity to be public for this peer review?** For information about this choice, including consent withdrawal, please see our Privacy Policy

Reviewer #1: No

Reviewer #2: **Yes:** Karen Alcos-Flores

---

## [Decision Letter · Decision Letter 1]

5 Mar 2026

Exploring the experiences of women of African, Caribbean and Mixed heritages to inform a music-based intervention for perinatal mental health in South East London: a qualitative study

PMEN-D-25-00439R1

Dear Miss Anstee,

We are pleased to inform you that your manuscript 'Exploring the experiences of women of African, Caribbean and Mixed heritages to inform a music-based intervention for perinatal mental health in South East London: a qualitative study' has been provisionally accepted for publication in PLOS Mental Health.

Best regards,

Gellan Karamallah Ramadan Ahmed

Academic Editor

PLOS Mental Health

Reviewer Comments (if any, and for reference):

Reviewer's Responses to Questions

**Comments to the Author**

Reviewer #1: All comments have been addressed

Reviewer #2: All comments have been addressed

publication criteria?

Reviewer #1: Yes

Reviewer #2: Yes

3. Has the statistical analysis been performed appropriately and rigorously?

Reviewer #1: N/A

Reviewer #2: Yes

4. Have the authors made all data underlying the findings in their manuscript fully available (please refer to the Data Availability Statement at the start of the manuscript PDF file)?

Reviewer #1: Yes

Reviewer #2: Yes

5. Is the manuscript presented in an intelligible fashion and written in standard English?

Reviewer #1: Yes

Reviewer #2: Yes

Reviewer #1: 1.1 Overview and general recommendation

Thank you very much for your work and for submitting an updated version of the manuscript. The manuscript is now aligned philosophically with your constructivist lens, goes deeper in the results section to engage more with the data and has addressed the minor comments throughout. I am satisfied that the comments have been actioned, and therefore I am happy to recommend the following article for publication.

Thank you so much for your hard work, and I look forward to seeing it published.

**Do you want your identity to be public for this peer review?** For information about this choice, including consent withdrawal, please see our Privacy Policy

Reviewer #1: No

Reviewer #2: **Yes:** Karen-Alcos
